

# Preoperative digital 6-minute walk test reveals risk of postoperative pulmonary complications in patients undergoing heart valve surgery: a pilot feasibility study

Lixuan Li[1,*], Yuqiang Wang[2,*], Zhengbo Zhang[1], Zeruxin Luo[3], Wenqing Wang[1], Jiachen Wang[4], Xiaoli Liu[1], Ying Shi[5], Tian Yuan[5], Yong Fan[1], Hong Liang[1], Yingqiang Guo[2], Buqing Wang[6], Jing Wang[7,8] and Jiaoxue Deng[7,8]

[1] Department of Medical Innovation and Research, Chinese PLA General Hospital, Beijing, China
[2] Department of Cardiovascular Surgery, Cardiovascular Surgery Research Laboratory and National Clinical Research Center for Geriatrics, West China Hospital, Sichuan University, Chengdu, China
[3] Department of Rehabilitation Medicine Center, West China Hospital, Sichuan University, Chengdu, China
[4] General Hospital of Tibet Military Region, Lhasa, China
[5] Chinese PLA Medical School, Beijing, China
[6] Department of Medical Engineering, Chinese PLA General Hospital, Beijing, China
[7] Beijing Key Lab of Traffic Data Analysis and Mining, School of Computer and Information Technology, Beijing Jiaotong University, Beijing, China
[8] CAAC Key Laboratory of Intelligent Passenger Service of Civil Aviation, Beijing, China
[*] These authors contributed equally to this work.

Corresponding authors
Jing Wang, wj@bjtu.edu.cn
Jiaoxue Deng,
jiaoxuedeng@bjtu.edu.cn

## ABSTRACT

**Background.** Postoperative pulmonary complications (PPCs) are a significant concern in cardiac surgery, affecting patient prognosis. This pilot study explored the feasibility of developing a machine learning model for preoperative PPCs risk stratification by integrating dynamic respiratory physiology from the six-minute walk test (6MWT) with clinical parameters.

**Methods.** A prospective study was conducted at the Department of Cardiovascular Surgery of West China Hospital, Sichuan University, from August 2021 to December 2022. We enrolled 142 consecutive patients undergoing valvular heart surgery. After quality control, 117 patients with complete synchronized respiratory monitoring during 6MWT and clinical data were included. We extracted 94 physiological features across 6MWT phases (baseline, walking, recovery) and clinical variables, developing predictive models using five machine learning algorithms evaluated through rigorous five-fold cross-validation.

**Results.** The logistic regression model demonstrated promising discriminative performance (AUC 0.86, 95% CI [0.81–0.89]) in this exploratory cohort. Preliminary physiological patterns emerged, including associations between elevated expiratory tidal volume during recovery (OR 9.70, $p = 0.006$) and reduced baseline minute ventilation (OR 0.15, $p = 0.002$) with higher PPCs risk.

**Conclusion.** These pilot findings suggest that continuous physiological monitoring during 6MWT, when combined with clinical data, may provide a feasible approach for preoperative PPCs risk assessment. While requiring multi-center validation, the

results highlight the potential of wearable-enabled respiratory monitoring to guide prehabilitation strategies in cardiac surgery.

## INTRODUCTION

Postoperative pulmonary complications (PPCs) are a substantial concern for approximately 10–25% of cardiac surgery patients (*Szelkowski et al., 2015*; *Hulzebos et al., 2006*), impairing their functional capacity and heightening mortality risks (*Ibañez et al., 2016*). This concern is particularly salient given the substantial population burden of valvular heart disease—over 40 million individuals worldwide suffer from mitral or aortic valve pathologies, with annual cardiac valve procedures exceeding 180,000 cases (*Davidson & Davidson, 2021*). Prehabilitation such as respiratory muscle strengthening can halve PPC incidences (*Boden & Denehy, 2022*), but most patients receive generalized rehabilitation packages, which are often expensive, time-consuming, and lack personalization for optimal efficacy (*Cheung & Chan, 2022*). Effective implementation requires precise preoperative risk stratification to identify candidates most likely to benefit. Therefore, conducting preoperative PPCs risk assessments for heart valve surgery patients is vital for guiding treatment plans and estimating healthcare resource requirements. This study focuses specifically on prehabilitation-oriented risk identification to inform targeted intervention strategies.

Several studies have constructed models/scores for risk assessment of PPCs. The most widely used PPCs risk assessment tool currently is the assess respiratory risk in surgical patients in Catalonia (ARISCAT), which was developed from a prospective multicenter large-sample study (*Canet et al., 2010*) and subsequently validated across multiple independent cohorts. A large European cohort (patients receiving general, neuraxial, or plexus block anesthesia) showed that the ARISCAT's discrimination was good, but performance differs between geographic areas (*Mazo et al., 2014*). ARISCAT also showed good performance in patients undergoing major emergency abdominal surgery at a Danish University Hospital (*Kokotovic et al., 2022*), elderly after upper abdominal surgery in Thailand (*Nithiuthai et al., 2021*), and patients after thoracic surgery (*Ülger et al., 2022*), upper and lower abdominal surgery (*Kara et al., 2020*) in Turkey, and renal transplant (*Kupeli et al., 2017*). Despite robust external validation, it was specifically designed for non-cardiac surgery and incorporates intraoperative factors, which limits its application in the preoperative setting. The Pulmonary Risk Score is a preoperative pulmonary risk assessment tool published in The Journal of the American Medical Association (JAMA) that determines a patient's probability of PPCs risk by scoring age, productive cough, diabetes mellitus, history of tobacco smoking, chronic obstructive pulmonary disease (COPD), body mass index, and pulmonary function tests (*Hulzebos et al., 2006*). The Pulmonary Risk Score was derived exclusively from a small coronary artery bypass grafting (CABG) cohort ($N = 106$) without external validation, constraining its broader application. *Khanna*

*et al. (2023)* developed and temporally validated a predictive model for post-cardiac surgery PPCs using a large cohort of 17,433 patients. However, the incorporation of intraoperative variables limits its preoperative applicability. Several additional large-scale PPC risk scores are similarly constrained in pre-cardiac surgery application, either due to their incorporation of intraoperative parameters or derivation from non-cardiac surgical cohorts (*Arozullah et al., 2000*; *Arozullah et al., 2001*; *Neto et al., 2018*; *Xue et al., 2021*). Moreover, existing PPC risk assessment models/scores are based on static data that only reflect the patient's situation at the moment of measurement.

The six-minute walk test (6MWT) is a submaximal exercise assessment tool commonly used to evaluate cardiorespiratory fitness and prognosis in patients, with the six-minute walk distance (6MWD) as the primary outcome indicator (*Agarwala & Salzman, 2020*). The 6MWD's natural subjectivity and static nature limit its application (*Costanzo et al., 2022*), particularly in PPCs risk stratification, with inconsistent research results and suboptimal performance metric (*Lee et al., 2020*; *Keeratichananont, Thanadetsuntorn & Keeratichananont, 2016*). Dynamic information during the 6MWT enables a multidimensional and effective assessment of a patient's functional status. In the digital age, wearable sensors can capture continuous physiological data during the 6MWT, such as electrocardiogram (ECG), respiration, blood pressure, blood oxygen levels, pulse, and acceleration, which offer great potential for personalized health assessments, disease monitoring, and early warnings through artificial intelligence-driven analysis, such as stratifying early cardiovascular risk *via* wearable devices (*Orini et al., 2023*). Dynamic physiological data during exercise, particularly respiratory system, offers insights into a patient's compensatory abilities and physiological status, which are closely linked to their overall physical condition. *Al-Khalidi et al. (2011)* showed that frequent changes in respiratory physiological parameters reflect impaired cardiorespiratory and neurological function. In patients with low exercise capacity, such as COPD or heart failure, increased breathing rates are compensatory responses to hypoxemia and hypercapnia (*Murata et al., 2020*; *Cretikos et al., 2008*). These findings collectively suggest that respiratory dynamics during 6MWT may serve as sensitive biomarkers for early identification of patients at heightened risk for postoperative pulmonary complications.

This pilot study investigates the feasibility of utilizing dynamic respiratory physiology data from the 6MWT, combined with clinical records, through machine learning algorithms to preoperatively identify the risk of PPCs in cardiac valve surgery patients, and determine key predictive risk factors.

# MATERIALS & METHODS

## Design

This prospective exploratory study was conducted at the Department of Cardiovascular Surgery of West China Hospital, Sichuan University, from August 2021 to December 2022. We enrolled a consecutive cohort of heart valve disease patients to investigate physiological responses during standardized 6MWT before surgery. The sample size reflected the available patient population during the study period, balancing the need for preliminary exploration

of 6MWT parameters with clinical feasibility. Healthcare professionals systematically collected continuous physiological monitoring data alongside routine preoperative clinical assessments.

A population of people who underwent cardiac surgical valve procedures was included. Patient exclusion criteria were: (1) emergency surgery; (2) fulfillment of American Thoracic Society (ATS) guidelines for 6MWT contraindications; (3) forced termination due to adverse events or other unforeseen circumstances; (4) non-pulmonary severe complications after surgery, such as gastrointestinal bleeding, cerebrovascular accidents, low cardiac output syndrome, and cardiac arrest; (5) unreadable or poor-quality signals from the wearable device. Written informed consent was obtained from all participants.

This study was approved by the Ethics Committee of the West China Hospital of Sichuan University, Ethics No. 20211023, Clinical Registration No. ChiCTR2100050005 (http://www.chictr.org.cn).

## Patients perioperative care information

Patients scheduled for elective cardiac valve surgery are either subjected to catheter-based interventions, such as transcatheter aortic valve replacement, or open-heart surgery. The surgical team, comprising a cardiologist, two cardiovascular surgeons, an echocardiographer, and an anesthesiologist, determines the surgical approach based on established cardiac valve surgery guidelines, patient preferences, and various other considerations. All elements of patient care, including preoperative preparation, prophylactic antibiotic administration, pain management, and general care, are coordinated by cardiac nurse specialists and physicians in accordance with standard clinical practice. Starting from the first postoperative day, all participants undergo early mobilization, chest physiotherapy, and additional physiotherapy services provided by the same experienced physiotherapy team in the intensive care unit and cardiology ward (*Luo et al., 2023*).

## Experimental procedure

Patients engaged in a 6MWT wearing a medical-grade wearable device (SensEcho®), which monitored physiological signals including ECG, $SpO_2$, respiratory, and triaxial acceleration (*Wang et al., 2022*), as shown in Fig. 1. SensEcho comprises a flexible vest with embedded fabric electrodes for single-lead ECG signal acquisition (200 Hz), alongside integrated sensor coils for chest/abdominal respiratory monitoring *via* respiratory inductive plethysmography (25 Hz). The accelerometer sensor integrated in the terminal collected posture/body movement signals (25 Hz). A ring worn on the thumb collected blood oxygen signals (1 Hz) in real-time and transmits them to the terminal *via* Bluetooth synchronization.

Patients wore SensEcho to continuously monitor physiological signals throughout the test. After a 1-minute rest, the test commenced in a 30-meter corridor adhering to ATS guidelines. Patients began walking from a start line, timed for 6 min, with a prompt on the device encouraging them at 1-minute intervals in a standardized way. Medical staff monitored ECG, respiration, and oxygen saturation on a portable android device and recorded the laps. The test was halted if symptoms such as chest pain, loss of consciousness,

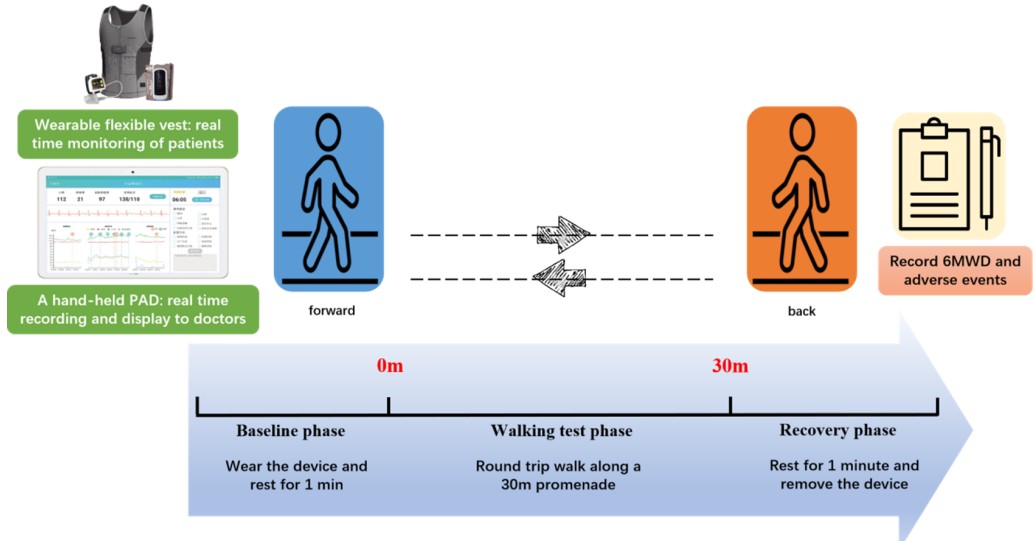

**Figure 1** Digital six-minute walking test.

severe dyspnea, falls, excessive sweating, or pallor occurred, indicating inability to complete the test unassisted. Participants could rest or terminate the test at any point. At the end of the walk the patient rests against the wall for 1 min.

## Outcomes

The main outcome, PPCs, were diagnosed in 14 days postoperatively according to the Melbourne Group Scale (MGS) (*Reeve et al., 2010*) as meeting at least four positive criteria (Table 1). The MGS was selected as our primary outcome measure based on robust validation data demonstrating its superior sensitivity for detecting severe PPCs, confirming its clinical utility for prehabilitation candidate selection (*Wang et al., 2023*). The secondary outcomes were ventilation failure, defined as mechanical ventilation lasting over 48 h, and pneumonia, diagnosed in accordance with the European Perioperative Clinical Outcomes criteria (*Jammer et al., 2015*).

## Sample size

This prospective study examines the predictive feasibility of continuous physiological monitoring for postoperative pulmonary complications in valve surgery patients using ROC curve analysis. We anticipate that the area under the ROC curve (AUC) will exceed 0.7, with the confidence interval for the AUC not exceeding 0.2, while accounting for a 5% dropout rate. Using PASS 15.0 software, a minimum sample size of 112 participants was required.

## Data pre-processing

The raw physiological signals were smoothed using a moving average filter. Outliers exceeding three standard deviations are identified and removed from the signal. Peaks and valleys were detected using Khodadad's method (*Khodadad et al., 2018*). Then,

**Table 1** Melbourne group scale evaluation criteria.

**At least the following four items can be determined to occur PPCs:**

1. Chest radiograph report of collapse/consolidation;

2. Leukocyte cell count $> 11.2 \times 10^9$/L or prescription of an antibiotic specific for respiratory infection (except for those routinely used after surgery);

3. Oral temperature $>38\ °C$, without fever caused by reasons other than lung;

4. Microbiological evidence of sputum (+);

5. Yellow or green sputum different from preoperative assessment;

6. $SpO_2$ is $<90\%$ in indoor environment;

7. Clinical diagnosis of pneumonia or pulmonary infection;

8. Stay in the care unit for $> 36$ h or enter the care unit again due to respiratory problems.

**Notes.**

PPCs, Postoperative pulmonary complications.

the physiological data quality was assessed by the respiratory signal quality assessment algorithms (*Xu et al., 2021*) combined with expert experience. Data exclusion criteria included equipment issues, detection errors of peaks and troughs, irregular waveforms, baseline drift, noise interference, and incomplete data collection.

## Feature extraction

The physiological data obtained from the wearable were divided into three segments according to time to extract physiological features for subsequent computational analyses: 1 min of rest before the start of the 6MWT is called base phase, 6 min of walking is called walk test phase, and 1 min of rest after the end is called recovery phase. Respiratory parameters, including breath rate, inspiratory/expiratory time, inspiratory time ratio, tidal volume, minute ventilation volume, abdominal-contribute, and labored breathing index were calculated for statistical features (mean, standard deviation, maximum, minimum, coefficient of variation). The acceleration time is the time required for the mean value of the base phase to rise to 75% of maximum value. The slope is the accelerated slope from the mean value of the base phase to 75% of the maximum value. Oxygen saturation ($SpO_2$) features included the value and desaturation in each phase. The calculation and definition of each parameter are shown in Table 2.

Clinical features were collected from the Cardiac Surgery Database by the investigators using a data collection form, including baseline demographics, 6MWD, New York Heart Association (NYHA) classification, preoperative laboratory tests, pulmonary function test, inspiratory muscle strength, comorbidities, medication use, and type of surgery. In this article, type of surgery refers to transcatheter aortic valve replacement (TAVR) and surgical aortic valve replacement (SAVR).

## Model

After data pre-processing and feature extraction, a logistic regression model was employed to refine a feature set by ranking features based on their predictive significance, where the absolute value of the model's coefficients determined feature importance (Fig. 2). The top-ranked features were selected to form an extracted feature set, and the optimal coefficient threshold (best threshold) was identified to enhance model performance. This

**Table 2  Definitions of physiological features.**

| Features | Definitions |
|---|---|
| **Breath Rate (BR)** | **Breaths per minute.** The means during base(BR_base), and recovery (BR_recovery) phases, the coefficient of variation(cv) during walk test phase, the maximum(max) during walk test phase(BR_max), the value at the end of walk test phase(BR_end), acceleration time and slope were extracted separately. |
| BRR1 | BR_end - BR_recovery |
| BR_increase | BR_max - BR_base |
| Time_20 | Duration of breath rate greater than 20 times/minute |
| **Inspiratory Time (TI)** | **Time required to complete an inhalation.** The means during base, walk test, and recovery phases and the coefficient of variation during walk test phase were extracted separately. |
| **Exhalation Time (TE)** | **Time required to complete an exhalation.** The means during base, walk test, and recovery phases and the coefficient of variation during walk test phase were extracted separately. |
| **Inspiratory Time ratio (TI_ratio)** | **Inspiratory time as a ratio of the respiratory cycle.** The means during base, walk test, and recovery phases and the coefficient of variation during walk test phase were extracted separately. |
| **Tidal Volume (VT)** | **Volume of air per inhalation or exhalation.** The means during base, walk test, and recovery phases and the coefficient of variation during walk test phase were extracted separately. |
| **Minute Ventilation Volume (MV)** | **Total air inhaled or exhaled per minute(Product of tidal volume and breath rate).** The means during base phases, the maximum during walk test phase, acceleration time and slope were extracted separately. |
| **Abdominal-contribute (AB_Contribute)** | **Abdominal volume as a ratio of tidal volume.** The means during base, walk test, and recovery phases and the coefficient of variation during walk test phase were extracted separately. |
| **Labored Breathing Index (LBI)** | **(Chest respiratory range of motion + Abdominal respiratory range of motion) / Total respiratory range of motion.** The means during base, walk test, and recovery phases and the coefficient of variation during walk test phase were extracted separately. |
| **Oxygen Saturation (SpO$_2$)** | The means during base, and recovery phases and the minimum during walk test phase were extracted separately. |
| SpO$_2$_6MWD | Mean oxygen saturation during walk test phase × six-minute walking distance |
| Desaturation | Area under the curve for oxygen saturation <90% during walking phase. |

threshold was iteratively refined until the final feature subset was established. Subsequently, the performance of several machine learning models—logistic regression (LR), support vector machine (SVM), linear discriminant analysis (LDA), K-nearest neighbor (KNN), and random forest classifier (RF)—was evaluated using stratified 5-fold cross-validation

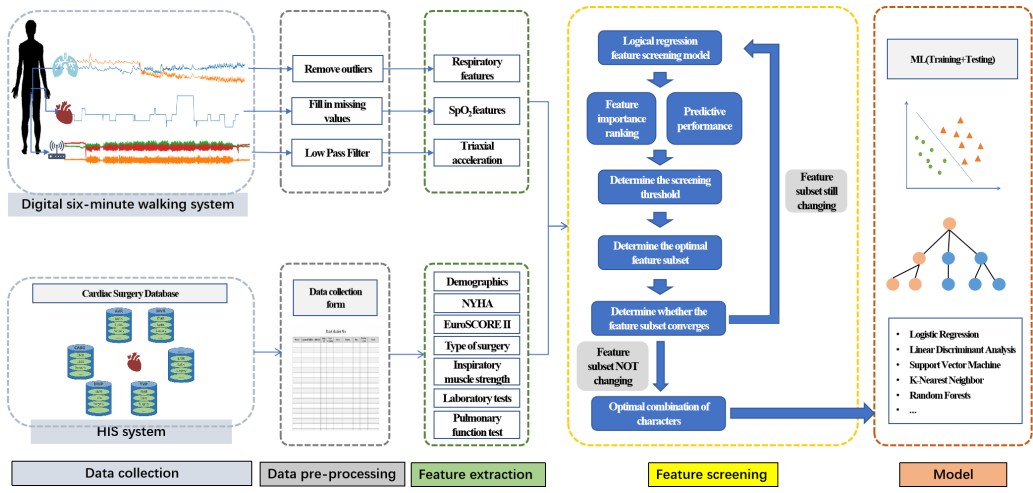

**Figure 2** Feature combination process.

on the selected feature set. Model evaluation metrics included AUC, sensitivity, specificity, accuracy, and F1 score. Continuous variables were expressed as mean ± standard deviation. Categorical variables were presented as frequencies and percentages (n, %). Between-group comparisons employed independent $t$-tests for normally distributed continuous variables and chi-square tests for categorical variables. All statistical analyses were conducted using Pytorch version 2.0.1.

# RESULTS

## Postoperative pulmonary complications

Among 142 eligible patients, 117 were involved in the analysis as they had physiological signals that met the quality requirements and no postoperative severe non-pulmonary complications (Fig. 3). No unexpected events or study withdrawals were reported throughout the study.

Among the study cohort, 26 patients (22.2%) developed postoperative pulmonary complications (PPCs group), while the remaining 91 patients (77.8%) had no pulmonary complications (non-PPCs group). PPCs patients were older with lower male predominance. Congestive heart failure was prevalent in both groups. PPCs patients exhibited significantly longer operative times (anesthesia/surgery) and greater blood loss across both surgical approaches. SAVR was more frequently performed in PPCs patients, who also experienced greater intraoperative blood loss, higher cardiopulmonary bypass circuit blood volumes, and increased autologous blood transfusion requirement during SAVR procedures.

The demographic and and intraoperative information are detailed in Tables 3 and 4, respectively. Table 3 demonstrates significant between-group differences in gender distribution ($p = 0.033$) and surgical method ($p < 0.001$). These two variables were consequently included as candidate predictors in our subsequent feature selection process. Table 4 demonstrates significant differences in intraoperative parameters between

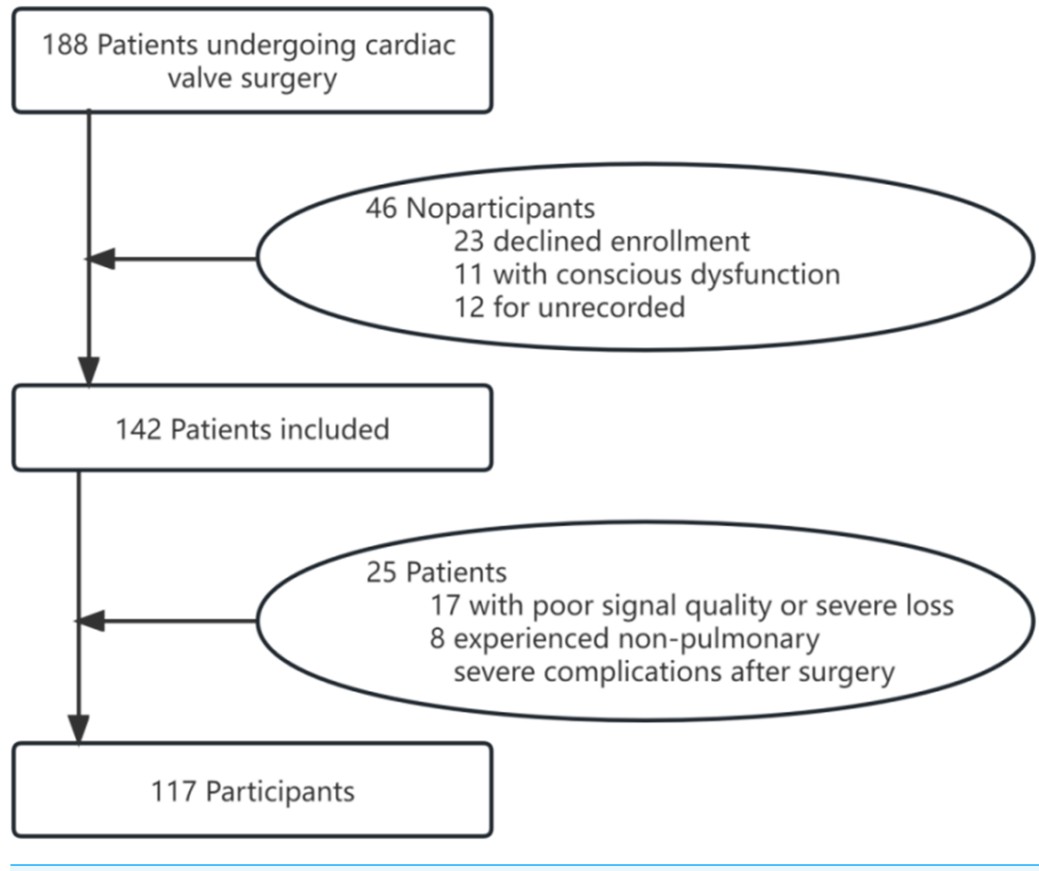

**Figure 3** Recruitment flowchart.

groups. While intraoperative factors are undeniably associated with PPC development, their inherent unpredictability during surgery limits optimization potential. Although preoperative parameters alone show weaker predictive value than comprehensive models incorporating perioperative intensive care unit data, this study deliberately focused on preoperatively modifiable factors, consistent with the concept of prehabilitation. Although intraoperative factors are critical, unforeseen events during surgery are inherently unpredictable and cannot be optimized. We therefore restricted our analysis to preoperative variables, excluding intraoperative factors that cannot be modified preemptively.

## Model performance

Through comprehensive feature extraction from multimodal data sources, we obtained a total of 94 parameters, including: 52 dynamic physiological variables from 6MWT and 42 parameters from the clinical information system. The optimal subset obtained by the feature screening model consists of three respiratory features: mean inspiratory minute ventilation during baseline stage (MV_in_base), maximum inspiratory minute ventilation during the walking phase (MV_in_max), mean expiratory tidal volume during recovery phase (VT_ex_re), and two clinical features: type of surgery and enhanced diuretic.

**Table 3  Demographic data.**

| Variable | Total ($N = 117$) | PPCs ($N = 26$) | Non-PPCs ($N = 91$) | P-value (PPCs *vs.* Non-PPCs) |
|---|---|---|---|---|
| **Demographics** | | | | |
| Gender (male), n(%) | 78 (66.7) | 12 (46.1) | 66 (72.5) | 0.023[*] |
| Age (years), mean ± SD | 65.18 ± 11.11 | 66.28 ± 10.98 | 61.59 ± 11.01 | 0.054 |
| Height (m), mean ± SD | 160.34 ± 8.11 | 161.01 ± 7.49 | 158.00 ± 9.77 | 0.155 |
| Weight (kg), mean ± SD | 60.68 ± 12.45 | 61.74 ± 10.64 | 56.94 ± 17.12 | 0.083 |
| BMI, mean ± SD | 23.47 ± 3.91 | 23.75 ± 3.29 | 22.48 ± 5.53 | 0.144 |
| NYHA classification, mean ± SD | 2.52 ± 0.53 | 2.48 ± 0.50 | 2.65 ± 0.63 | 0.153 |
| EuroSCORE II, mean ± SD | 4.85 ± 2.82 | 4.98 ± 2.82 | 4.42 ± 2.83 | 0.379 |
| 6MWD, mean ± SD | 395.98 ± 84.21 | 396.18 ± 88.40 | 395.31 ± 69.08 | 0.963 |
| **Comorbidities, n (%)** | | | | |
| Hypertension | 49 (41.8) | 9 (34.6) | 40 (43.9) | 0.531 |
| Asthma | 2 (1.7) | 0 | 2 (2.1) | 0.446 |
| Preoperative smoking history | 32 (27.3) | 5 (19.2) | 27 (29.6) | 0.422 |
| Preoperative anemia | 4 (3.4) | 1 (3.8) | 3 (3.2) | 0.892 |
| Respiratory infection in the past month | 4 (3.4) | 2 (7.6) | 2 (2.1) | 0.455 |
| Congestive heart failure | 97 (82.9) | 20 (76.9) | 77 (84.6) | 0.533 |
| Preoperative hypoxemia | 46 (39.3) | 12 (46.2) | 34 (37.3) | 0.561 |
| Previous thoracotomy | 6 (5.1) | 1 (3.8) | 5 (5.4) | 0.737 |
| **Pulmonary function test, mean±SD** | | | | |
| FEV1 (ml) | 1,991.69 ± 626.76 | 1,897.66 ± 677.61 | 2,018.55 ± 612.76 | 0.388 |
| FEV1-predicted (%) | 78.17 ± 22.52 | 71.03 ± 18.19 | 80.20 ± 23.30 | 0.067 |
| FVC (ml) | 2,523.33 ± 765.03 | 2,410.22 ± 915.43 | 2,555.65 ± 718.90 | 0.395 |
| FVC-predicted (%) | 75.58 ± 19.55 | 70.22 ± 17.98 | 77.11 ± 19.81 | 0.114 |
| MVV (ml) | 63.01 ± 24.09 | 58.77 ± 28.87 | 64.22 ± 22.58 | 0.311 |
| MIP (cmH$_2$O) | 57.26 ± 19.06 | 56.31 ± 23.90 | 57.53 ± 17.58 | 0.810 |
| MIP-predicted (%) | 66.94 ± 18.12 | 63.62 ± 17.68 | 67.89 ± 18.22 | 0.291 |
| FEV1/FVC (%) | 79.51 ± 13.09 | 79.62 ± 8.43 | 79.48 ± 14.18 | 0.963 |
| Surgical method (SAVR), n (%) | 54 (46.1) | 20 (76.9) | 34 (37.3) | 0.000[*] |

**Notes.**

BMI, Body Mass Index; NYHA, New York Heart Association; 6MWD, six-minute walk distance; PPCs, Postoperative pulmonary complications; FEV1, Forced expiratory volume in 1 s; FVC, Forced vital capacity; MVV, Maximum Ventilatory Volume; MIP, Maximum Inspiration Pressure; TAVR, Transcatheter Aortic Valve Replacement; SAVR, Surgical Aortic Valve Replacement.

[*]$p < 0.05$.

Different machine learning models, including LR, SVM, LDA, RF, and KNN, were implemented using the selected feature subset. Table 5 contrasts the predictive performance of different machine learning models. LR achieved superior discrimination (AUC 0.86 ± 0.07, 95% CI [0.81–0.89]), closely followed by LDA (AUC 0.85 ± 0.09, 95% CI [0.79–0.88]). This consistent performance across linear models suggests that the select feature subset exhibits robust discriminative ability across distinct classification methodologies. While our analysis revealed consistent underperformance of non-linear models (SVM, RF, KNN) compared to linear classifiers, these findings require validation in larger cohorts. Figure 4 compares the receiver operating characteristic (ROC) curves of the different models, which depicts the average results of the 5-fold cross-validation.

**Table 4  Intraoperative information.**

| Variable | | PPCs ($N = 26$) | Non-PPCs ($N = 91$) | Total ($N = 117$) |
|---|---|---|---|---|
| Anesthesia time (min) | SAVR | $356.75 \pm 16.69$ | $268.85 \pm 12.96$ | $477.50 \pm 32.13$ |
| | TAVR | $171.50 \pm 38.98$ | $92.83 \pm 30.16$ | $32.50 \pm 15.15$ |
| Surgery time (min) | SAVR | $349.00 \pm 11.57$ | $263.12 \pm 11.18$ | $388.24 \pm 20.69$ |
| | TAVR | $132.75 \pm 4.1$ | $69.53 \pm 3.4$ | $25.7 \pm 3.34$ |
| Blood loss (ml) | SAVR | $351.87 \pm 9.47$ | $265.24 \pm 8.46$ | $421.30 \pm 18.44$ |
| | TAVR | $136.44 \pm 5.24$ | $71.75 \pm 4.14$ | $26.35 \pm 3.30$ |
| CPB circuit blood (ml)-SAVR | | $419.00 \pm 34.38$ | $379.41 \pm 21.02$ | $394.07 \pm 18.36$ |
| Autologous blood (ml)-SAVR | | $377.50 \pm 30.67$ | $275.00 \pm 22.06$ | $312.96 \pm 19.03$ |
| CPB time (min)-SAVR | | $125.15 \pm 9.47$ | $120.50 \pm 6.89$ | $122.22 \pm 5.53$ |
| Aortic cross-clamp time (min)-SAVR | | $85.85 \pm 6.12$ | $85.79 \pm 5.05$ | $85.81 \pm 3.87$ |

**Notes.**

TAVR, Transcatheter Aortic Valve Replacement; SAVR, Surgical Aortic Valve Replacement; CPB, Cardiopulmonary bypass.

**Table 5  Performance of different machine learning models.**

| | AUC (95% CI) | ACC | F 1 | Sensitivity | Specificity |
|---|---|---|---|---|---|
| Logistic regression | **0.86 $\pm$ 0.07 (0.81–0.89)** | **0.83 $\pm$ 0.03** | 0.84 $\pm$ 0.04 | **0.74 $\pm$ 0.22** | **0.86 $\pm$ 0.29** |
| Linear discriminant analysis | 0.85 $\pm$ 0.09 (0.79–0.88) | 0.82 $\pm$ 0.03 | 0.85 $\pm$ 0.04 | 0.71 $\pm$ 0.19 | **0.86 $\pm$ 0.05** |
| Support Vector Machine | 0.71 $\pm$ 0.10 (0.64–0.76) | 0.72 $\pm$ 0.12 | 0.74 $\pm$ 0.14 | 0.49 $\pm$ 0.43 | 0.79 $\pm$ 0.04 |
| K-Nearest Neighbor | 0.61 $\pm$ 0.09 (0.56–0.67) | 0.78 $\pm$ 0.01 | **0.88 $\pm$ 0.01** | 0.20 $\pm$ 0.18 | 0.78 $\pm$ 0.01 |
| Random Forests | 0.63 $\pm$ 0.12 (0.52–0.71) | 0.74 $\pm$ 0.05 | 0.77 $\pm$ 0.06 | 0.34 $\pm$ 0.21 | 0.80 $\pm$ 0.03 |

**Notes.**

The bold values highlight the best-performing model for each evaluation metric.

Figure 5 shows the importance ranking of the five selected features based on shapley additive explanation (SHAP) values.

In addition, we used logistic regression to predict ventilation failure and pneumonia and obtained good predictive performance with AUC of 0.82, respectively, as detailed in Table 6.

The final logistic regression model for predicting postoperative pulmonary complications (PPCs) is presented in Table 7. The results revealed that enhanced diuretic emerged as the most significant risk factor for PPCs (OR = 10.64, $P = 0.007$), while patients undergoing SAVR showed a 6-fold higher risk compared to TAVR (OR = 6.01, $P = 0.003$). Among the 6MWT-derived physiological parameters, VT_ex_re exhibited a strong protective effect (OR = 9.70, $P = 0.006$), and MV_in_base showed a significant inverse association with PPCs risk (OR = 0.15, $P = 0.002$). Although MV_in_max did not reach conventional statistical significance ($p = 0.156$), we retained this variable in the final model as its exclusion led to a reduction in model AUC, indicating meaningful contribution to predictive accuracy, and its negative coefficient direction aligned with physiological expectations that patients with greater ventilatory reserve may be less susceptible to PPCs development.

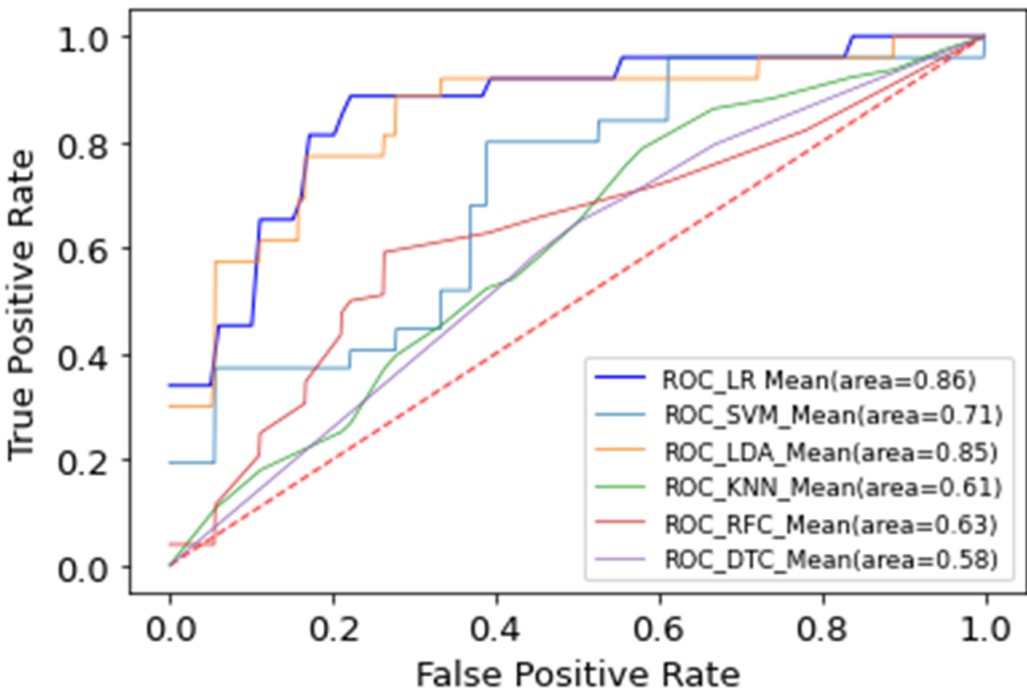

**Figure 4   Multiple machine learning model performance.**

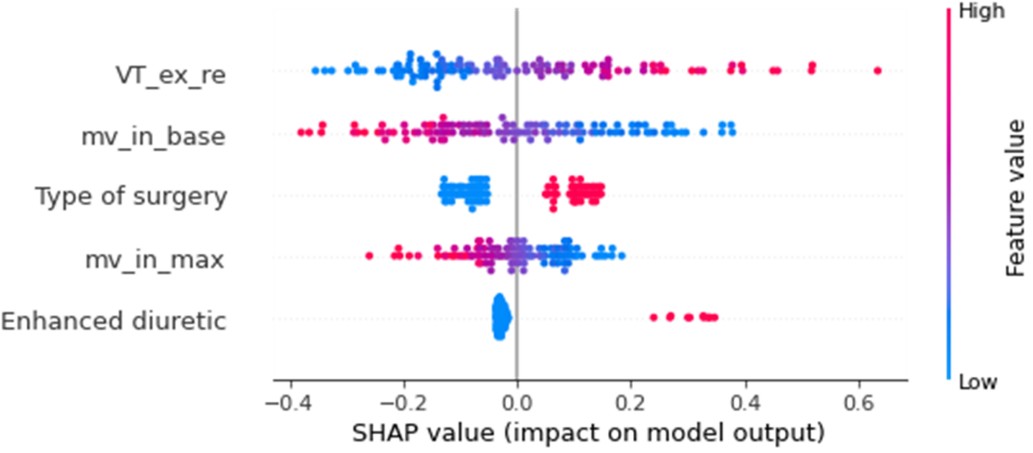

**Figure 5   Shapley value of extracted features.** VT_ex_re, mean expiratory tidal volume during recovery phase; MV_in_base, mean inspiratory minute ventilation during baseline stage; MV_in_max, maximum inspiratory minute ventilation during the walking phase.

**Table 6  Combining physiological and clinical data to predict single PPC outcomes.**

| PPC outcome | AUC (95% CI) | ACC | F1_score | Sensitivity | Specificity |
|---|---|---|---|---|---|
| Ventilation failure | 0.82 ± 0.18 (0.71–0.93) | 0.87 ± 0.05 | 0.88 ± 0.07 | 0.20 ± 0.04 | 0.93 ± 0.02 |
| Pneumonia | 0.82 ± 0.12 (0.73–0.88) | 0.80 ± 0.07 | 0.80 ± 0.08 | 0.78 ± 0.19 | 0.83 ± 0.03 |

Notes.
  PPC, Postoperative pulmonary complication.

**Table 7  The logistic regression model for identifying PPCs risk before cardiac valve surgery.**

| Variable | Coefficient | Odds ratio | 95% CI lower | 95% CI upper | *P*-value |
|---|---|---|---|---|---|
| Type of surgery | 1.79 | 6.01 | 0.61 | 2.97 | 0.003[*] |
| Enhanced diuretic | 2.36 | 10.64 | 0.66 | 4.07 | 0.007[*] |
| MV_in_base | −1.92 | 0.15 | −3.14 | −0.69 | 0.002[*] |
| MV_in_max | −1.00 | 0.37 | −2.39 | 0.38 | 0.156 |
| VT_ex_re | 2.27 | 9.70 | 0.65 | 3.90 | 0.006[*] |

Notes.
  MV_in_base, mean inspiratory minute ventilation during baseline stage; MV_in_max, maximum inspiratory minute ventilation during the walking phase; VT_ex_re, mean expiratory tidal volume during recovery phase.
  [*]$p < 0.05$.

# DISCUSSION

This pilot study investigated the feasibility of using 6MWT derived respiratory parameters to preoperatively predict PPCs and identify characteristic risk patterns in patients undergoing cardiac valve surgery. The developed model revealed clinically meaningful ventilatory abnormalities that may serve as preoperative warning signs for PPCs.

PPCs occurrence is linked to high-cost medical events, increased hospital mortality, and poorer long-term quality of life (*Ibañez et al., 2016*). Our study found a PPCs incidence of 22.22%, aligning with previous reports (21.9%) (*Wang et al., 2023*). For patients undergoing cardiac surgery, multiple factors including diaphragmatic hypotonia, decreased functional residual capacity, ventilator-induced lung injury, and surgical maneuvers impaired lung function (*Tanner & Colvin, 2020*), highlighting the need for improved preoperative risk stratification in resource-conscious healthcare systems.

Despite the widespread use and acceptance of the 6MWT for cardiopulmonary assessment, 6MWD may be insufficient for reliable risk stratification of PPCs in surgical populations (*Lee et al., 2020*; *Keeratichananont, Thanadetsuntorn & Keeratichananont, 2016*). This underscores the importance of incorporating dynamic physiological parameters during the test. Our model achieved superior discrimination (AUC 0.86 ± 0.07, 95% CI [0.81–0.89]) by integrating respiratory patterns with clinical data. This integration has also demonstrated a notable ability in the independent prediction of ventilation failure and pneumonia, with AUC of 0.82 for both. These findings foreshadow the potential value of continuous physiological data for long-range longitudinal prediction, which may aid clinicians in developing personalized treatment plans and early disease detection using wearable digital health technology (*Friend, Ginsburg & Picard, 2023*).

The model identified five risk factors for PPCs, including MV_in_base, MV_in_max, VT_ex_re, type of surgery, and enhanced diuretic. Our analyses suggested potential

associations between specific ventilatory parameters and PPCs risk, with preliminary evidence indicating that higher expiratory tidal volumes during recovery phases and lower minute ventilation at baseline may represent clinically relevant breathing patterns worthy of further investigation. VT_ex_re demonstrated a particularly strong predictive value (OR = 9.70, $p = 0.006$), while MV_in_base exhibited a protective effect (OR = 0.15, $p = 0.002$). Inefficient ventilation and limited oxygen supply are issues during quiet breathing in patients with cardiopulmonary dysfunction. Lung dysfunction leads to significant breathlessness and hypoxia intolerance, prompting shallow breathing as a coping mechanism (*Killian, Bucens & Campbell, 1982*). The inability of patients to achieve a specific minute ventilation while breathing is not due to being 'incapable' but rather because the resulting dyspnoea makes them 'unwilling' to do so (*Shea et al., 1989*). Accordingly, a lower minute ventilation during rest may signal a higher PPC risk. High tidal volumes after recovery suggest poorer cardiorespiratory fitness. High tidal volumes during recovery reflect poorer cardiorespiratory fitness. Tidal volume increases during walking. Ventilation is inefficient in patients with cardiopulmonary dysfunction, resulting in more $CO_2$ retention that cannot be eliminated in a timely manner. The body tries to match alveolar ventilation to carbon dioxide production to keep $PaCO_2$ stable (*Ohashi et al., 2013*). During impulsive exercise, rest-work/work-rest transitions, and intermittent exercise, tidal volume exhibits delayed responsiveness to sudden changes in workload rates (*Nicolò et al., 2017*). This aligns with our research findings: tidal volume gradually increases during walking exercises, maintaining a certain level even in the recovery phase after ceasing walking, necessary for expelling excessive $CO_2$ generated during the exercise. In contrast to individuals with efficient ventilation, those with lower ventilation efficiency retained more $CO_2$ under the same exercise load, necessitating a longer duration in the recovery phase and an increase in tidal volume to achieve a steady $PaCO_2$ (*Nicolò & Sacchetti, 2023*). Therefore, the observed physiological patterns in this pilot study could reflect underlying cardiorespiratory impairments that might contribute to PPC susceptibility, though additional validation is needed to establish definitive causal relationships.

Clinical parameters notably surgery type (OR = 9.70, $p = 0.003$) and enhanced diuretic use (OR = 10.64, $p = 0.007$) were strong predictors. Surgery-related factors pose an immutable risk to PPCs with TAVR and SAVR differing in their site and approach. According to the survey, the proportion of patients who developed PPCs during hospitalization for peripheral, abdominal, and intrathoracic surgery was 5.8%, 23.0%, and 51.3%, respectively (*Holland et al., 2014*). SAVR is more traumatic during hospitalization and results in a higher rate of PPCs than TAVR, consistent with our findings. Heart failure is a high risk factor for PPCs (*Fernandez-Bustamante et al., 2017*), and guidelines recommend that surgery in patients with acute or chronic heart failure should ideally be postponed until the patient is treated and cardiac function improves (*Kristensen et al., 2014*). Consistent with our observations, patients receiving intensive anti-heart failure therapy for severe heart failure prior to surgery are at higher risk of PPCs.

It is important to emphasize that our 6MWT-based model is designed specifically for preoperative risk stratification to guide prehabilitation decisions, rather than to replace comprehensive perioperative risk assessment tools. Traditional scoring systems

incorporating preoperative, intraoperative and intensive care unit parameters remain essential for postoperative risk prediction, as they capture a broader range of determinants including surgical complexity, anesthesia duration, and early postoperative course (*Khanna et al., 2023*). Our model complements these tools by identifying modifiable preoperative risk factors that can be targeted through prehabilitation interventions, while traditional scores maintain their value for postoperative monitoring and management.

While this study provides valuable insights into preoperative PPC prediction, several limitations should be acknowledged. As an exploratory analysis, this study was not powered for definitive clinical conclusions but rather to identify promising physiological markers for future investigation. First, the sample size ($n = 117$) and exploratory nature of this investigation necessitate cautious interpretation of the results, with validation in larger multicenter cohorts being essential to confirm these preliminary findings. Second, our analysis included patients undergoing two distinct surgical procedures (SAVR and TAVR) with inherently different PPC risk profiles. While we accounted for this variability by incorporating surgical approach as a model input variable, the resultant heterogeneity— though potentially enhancing generalizability across valve surgery populations—introduces clinical implementation considerations that merit attention. Third, the study did not develop a clinically practical scoring system, as appropriate score derivation would require substantially larger sample sizes to ensure robust cutoff values and risk stratification. Instead, we focused on establishing proof-of-concept for the predictive value of dynamic physiological parameters. Fourth, while we identified potentially important respiratory patterns, the observational nature of this study cannot establish causal relationships between these physiological markers and PPC development.

Future directions should include: (1) expansion to multi-center studies with adequate power for clinical score development; (2) longitudinal investigation of physiological patterns in valve disease patients to establish quantitative health status indicators; and (3) translation of the model into clinical decision support tools (*e.g.*, risk calculation software) that could integrate with electronic health records to facilitate personalized prehabilitation planning. Importantly, any clinical implementation should complement rather than replace comprehensive perioperative assessment tools that incorporate intraoperative and postoperative factors.

These limitations notwithstanding, the current findings provide a foundation for developing more precise preoperative risk assessment approaches that could ultimately guide targeted prehabilitation strategies for high-risk patients.

## CONCLUSIONS

Continuous physiological signals is individual "big data" reflecting health status. This pilot study provides preliminary evidence that continuous physiological monitoring during 6MWT may offer valuable insights for preoperative PPC risk assessment in cardiac valve surgery patients. Our exploratory analysis identified potentially important physiological markers that warrant further investigation. While these findings suggest the promise of wearable-enabled physiological monitoring for guiding prehabilitation strategies, the

moderate sample size and single-center design necessitate validation in larger, multicenter cohorts before clinical implementation. The study establishes a foundation for future research to refine risk prediction models, and develop frameworks for incorporating continuous physiological data into perioperative care pathways. These results highlight both the potential of personalized physiological profiling for surgical risk assessment and the importance of additional validation to confirm these initial observations. It heralds the widespread clinical application of continuous physiological monitoring using wearable devices.

## ACKNOWLEDGEMENTS

The authors would like to thank all the volunteers for participating in this work.

### Funding
This work was funded by the Natural Science Foundation of China (62171471) and National Clinical Research Center for Geriatrics, West China Hospital, Sichuan University (Z2024YY001). The funders had no role in study design, data collection and analysis, decision to publish, or preparation of the manuscript.

### Grant Disclosures
The following grant information was disclosed by the authors:
The Natural Science Foundation of China: 62171471.
National Clinical Research Center for Geriatrics, West China Hospital, Sichuan University: Z2024YY001.

### Competing Interests
The authors declare there are no competing interests.

### Author Contributions

- Lixuan Li analyzed the data, prepared figures and/or tables, authored or reviewed drafts of the article, and approved the final draft.
- Yuqiang Wang conceived and designed the experiments, performed the experiments, prepared figures and/or tables, authored or reviewed drafts of the article, and approved the final draft.
- Zhengbo Zhang conceived and designed the experiments, analyzed the data, prepared figures and/or tables, authored or reviewed drafts of the article, and approved the final draft.
- Zeruxin Luo conceived and designed the experiments, performed the experiments, authored or reviewed drafts of the article, and approved the final draft.
- Wenqing Wang analyzed the data, authored or reviewed drafts of the article, and approved the final draft.
- Jiachen Wang analyzed the data, authored or reviewed drafts of the article, and approved the final draft.

- Xiaoli Liu analyzed the data, authored or reviewed drafts of the article, and approved the final draft.
- Ying Shi analyzed the data, authored or reviewed drafts of the article, and approved the final draft.
- Tian Yuan analyzed the data, authored or reviewed drafts of the article, and approved the final draft.
- Yong Fan analyzed the data, authored or reviewed drafts of the article, and approved the final draft.
- Hong Liang analyzed the data, authored or reviewed drafts of the article, and approved the final draft.
- Yingqiang Guo conceived and designed the experiments, authored or reviewed drafts of the article, and approved the final draft.
- Buqing Wang analyzed the data, authored or reviewed drafts of the article, and approved the final draft.
- Jing Wang conceived and designed the experiments, analyzed the data, authored or reviewed drafts of the article, and approved the final draft.
- Jiaoxue Deng analyzed the data, authored or reviewed drafts of the article, and approved the final draft.

## Human Ethics

The following information was supplied relating to ethical approvals (i.e., approving body and any reference numbers):

This study was approved by the Ethics Committee of the West China Hospital of Sichuan University (Ethics No. 20211023).

## Clinical Trial Ethics

The following information was supplied relating to ethical approvals (i.e., approving body and any reference numbers):

The clinical trial described in this article was registered at the Chinese Clinical Trial Registry (no. ChiCTR2100050005).

## Data Availability

The raw data are available in the Supplemental File.

## Clinical Trial Registration

The following information was supplied regarding Clinical Trial registration:

The Chinese Clinical Trial Registry: no. ChiCTR2100050005.

## Supplemental Information

Supplemental information for this article can be found online at http://dx.doi.org/10.7717/peerj.19732#supplemental-information.

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
