# Peer review of "Preoperative digital 6-minute walk test reveals risk of postoperative pulmonary complications in patients undergoing heart valve surgery: a pilot feasibility study"

_PeerJ, doi:10.7717/peerj.19732_

## Round 0.1 · original submission · Major Revisions

Please respond to the reviewers comments, point by point

Reviewer 1 ·

Basic reporting

The authors should be commended on an interesting comparison of traditional measures for identifying PPC and the use of physiological data derived from the 6 minute walk test.

The language is clear and unambiguous

I have significant reservations regarding two major identified deficiencies in the study design, or that they have not been clearly communicated in the introduction.

1) The main outcome is PPC by the Melbourne Group Scale and was compared to physiological data obtained from the 6MWT, a post-hoc ARISCAT or PRS. I am unclear if these outcomes are similar, nor if the baseline comparison of these measures are equivalent. For example, the ARISCAT measures for the presence of multiple respiratory complications, including atelectasis, pneumonia. The Melbourne Group Scale mimics the NIH pneumonia criteria, and represents the presence of pneumonia. This varied assessment of outcome measures is difficult to compare, as these scales measure different outcomes. Similarly, the PRS was devised to assess a wide variety of PPC. This is problematic and the authors should more clearly define their study design and hypothesis to ensure that the comparators are comparing similar endpoints. PPCs are particularly challenging because of the wide variety of endpoints, frequently differing from study to study. This highlights the challenges of devising study examining strategies to reduce their occurrence. I certainly agree with the clinical findings of 6MWT variables being more 'predictive' of pneumonia, however, the above constitutes a critical flaw that needs to examined.

Experimental design

Although cross validation results appear to be helpful in prediction, this finding needs substantial validation in larger clinical cohort prior to determining whether the suggested tool is valuable for prediction of pneumonia. Improved and clear discussion of the variables inputted in the models are needed. Improved CV confidence intervals with discussion is needed. I want to reiterate that comparing the above referenced tools is a fatal flaw of this study. This should be re-written as a pilot study of physiological data + comorbidities +procedural variability, assessed with machine learning to develop a predictive model from 94 variables. A power analysis is conspicuously absent. If PPC is defined as pneumonia and there is approximately a 5% pneumonia risk after cardiac surgery, this would suggest that this sample size is inadequate for the discussed conclusions.

Validity of the findings

See above.

Additional comments

Although the authors should be highly commended on their study, the methodological flaws suggest that rejection with re-submission is warranted. The authors need to clean up the hypothesis, develop a clear design, and test their design with a validation cohort or suggest methods to do so. The conclusions should be toned down given the low sample size. Clearer language regarding the quality of model stability should be considered. A clinically relevant scoring system converted from the model logistic regression equation should be considered. As written, the authors do not meet the criteria that they set forth regarding their hypothesis.

Reviewer 2 ·

Basic reporting

No Comment

Experimental design

The current manuscript aims to reveal the risk of postoperative pulmonary complications in patients undergoing heart valve surgery by performing preoperative digital 6-minute walk test. I appreciate the authors for conducting the study in order to predict the postoperative pulmonary complications of heart valve surgery. However, the following concerns should be addressed.

Comments:
Among 117 participants actively involved with the study, please list the differences observed in case of gender, age, fasting status, body mass, and patients with other additional health complications. Please list the sample size according to the above-mentioned criteria and provide the threshold differences observed.

Validity of the findings

No Comment

Reviewer 3 ·

Basic reporting

The investigators should be congratulated for testing the 6 minute walk test (6MWT) before cardiac interventions (valve replacement or TAVR). Besides information obtained from patient’s history, a dynamic functional test such as the 6MWT could allow better discrimination of patients who will exhibit PPCs. Although the predictive performance of a model based on data extracted from the 6MWT was shown superior to the Pulmonary Risk Score (PRS) and the ARISCAT score, these results are not convincing and are fraud with major drawbacks given issues regarding the overall scope of the study and the methodology used.
The chosen traditional risk scores are inappropriate in cardiac surgery (see paragraph below Introduction). Not surprisingly, the AUCs of the ARISCAT and PRS was very low. It would have been more suitable to select a whole battery of preoperative and intraoperative data (e.g., MET or DASY, diabetes, HT, severity of heart failure and COPD or restrictive lung diseases, NYHA grade, anemia based on Hb, renal dysfunction based on BUN/Creat, liver dysfunction based on ASAT/ALAT, bilirubin, albumin, duration of surgery, transfusion,…).
The study population include patients with valvular aortic stenosis who undergo two different procedures (TAVR and surgical valve replacement) with different physiological consequences on the respiratory system. Such heterogeneity in patient population precludes further analysis. Moreover the number of patient is too small to test and subsequently validate the proposed new scoring model.

The content of the Introduction appears a little too vague and needs to be addressed more precisely and there are several inaccurate statements.
For instance, the terms “preoperative prehabilitation” is confusing and inadequate. By definition prehabilitation refers to all preoperative interventions involving identification of fixed/reversible risk factors, optimization of medical treatments, enhancement of physical fitness, psychological support and correction of nutritional defects.
The Pulmonary Risk Score (PRS) was used by Huzelbos in their study published in JAMA. The PRS was established based on analysis of a small cohort of patients undergoing CABGS (N=106) in 2003-6. The ARISCAT scoring system has been validated in several populations to assess the risk of PPCs after noncardiac surgery but not cardiac surgery.
So far, the most exhaustive and validated score to predict PPCs after cardiac surgery has been published by Khanna AK et al in 2023 (JCVTS 2023 A nomogram to predict postoperative pulmonary complications after cardiothoracic surgery) based on analysis of a large cohort including 17’433 patients.
The 6MWT is indeed a dynamic test to assess patient’s functional activity and it is best suited for patient with chronic lung diseases. In patients with chronic heart failure, the 6MWT was tested by Usko-Lencer et al in a cohort of 337 patients. However, there are several limitations for using the 6MWT before cardiac surgery since it is impossible or difficult to perform in patients with osteoarticular and musculoskeletal problems and it is contraindicated in patients who are in critical conditions (acute coronary syndrome, acute heart failure, aortic dissection,…).

Experimental design

Information regarding the conduct of anesthesia (IV, inhaled anesthetics, neuromuscular agents and reversing agents, blood transfusion, amount of clear fluids,…), surgery (valve replacement/repair, combined valve and coronary surgery) and cardiopulmonary bypass (duration of CPB and aortic cross-clamping), should be provided.
Information regarding patient selection should be provided (surgical valve and TAVR).
Selecting the Melbourne Scale Evaluation criteria is of limited value to address the full scope of PPCs after cardiac surgery (pleural effusion is not recorded, no grade of severity based on treatment intensity).
To make the tested scoring valid and reliable, there should be few/no exclusion criteria except contraindications to the test or inability, adverse events during the test. Therefore, it appears inappropriate to exclude a priori patients who will present non-pulmonary postoperative complications, some of whom may also present PPCs.
Power analysis and sample size calculation are lacking.

Validity of the findings

Much information interesting for clinicinas involved in perioperative care is lacking

---

## Round 0.2 · Minor Revisions

Please address these changes and resubmit.

Reviewer 1 ·

Basic reporting

Expansion of the study procedures for replication.

Experimental design

Adequately improved wording to reflect exploratory nature of the study.

Validity of the findings

Exploratory in nature, needs to provide justification for pre-planning sample size selection.

Additional comments

The authors should be commended on their thoughtful revisions. The manuscript better reports and represents the findings. I have some minor requests that should improve the manuscript:

1. The procedural details do not permit adequate replication for other researchers. Please expand the study procedures to include greater detail of: the device manufacturer and name, the timing and frequency of vital signs, the objective criteria by which participants were included in the study.
2. Reduce the number of nonstandard acronyms, as it is frustrating to have to go back and find the definition throughout the manuscript.
3. Please clarify this statement: Patients with enhanced diuretic use preoperatively are at a higher PPCs risk, aligning with guidelines delaying surgery until heart function improves.
4. Please provide some justification for the selected sample size, based on your pre-planning

Thank you

---

## Round 0.3 · accepted · Accept

All the comments were addressed.

Reviewer 1 ·

Basic reporting

No concerns

Experimental design

No concerns

Validity of the findings

No concerns

Additional comments

No concerns

Reviewer 2 ·

Basic reporting

no comment

Experimental design

no comment

Validity of the findings

no comment